# How Do Socio-Demographic Factors, Health Status, and Knowledge Influence the Acceptability of Probiotics Products in Hong Kong?

**DOI:** 10.3390/foods13182971

**Published:** 2024-09-19

**Authors:** Zilin Xu, Nan Wu, Shun Wan Chan

**Affiliations:** 1School of Biological Sciences, Faculty of Science, The University of Hong Kong, Hong Kong, China; zilin.xu@connect.polyu.hk (Z.X.); u3604602@connect.hku.hk (N.W.); 2Department of Food Science and Nutrition, The Hong Kong Polytechnic University, Hong Kong, China; 3Department of Food and Health Sciences, Technological and Higher Education Institute of Hong Kong, Hong Kong, China

**Keywords:** probiotics, socio-demographics, health status, knowledge

## Abstract

In recent years, due to growing interest in gut health, the potential benefits of probiotics on the gut have received much attention. Probiotics, now readily available in both dietary supplements and a variety of foods, have become a focal point of consumer health choices. This study aims to explore the impact of consumer-related factors, including socio-demographic profiles, health status, and probiotics knowledge, on the acceptance of probiotics products in Hong Kong. A total of 385 participants engaged in a survey, providing data for an in-depth analysis of how these factors influence attitudes toward probiotics. Findings revealed a general confidence in the safety of probiotics products among respondents; however, there was a noticeable gap in probiotics understanding. The study highlighted a correlation between probiotics knowledge and specific socio-demographic attributes, with higher educational attainment positively linked to greater probiotics awareness. Furthermore, the research indicated that women exhibit higher health consciousness and a greater propensity for probiotics consumption compared to men. Consequently, promoting enhanced probiotics education and fostering increased health awareness are crucial steps to prevent the misuse of probiotics and optimize health outcomes.

## 1. Introduction

As public awareness of health and nutrition has increased in recent years, particularly after the COVID-19 pandemic, there has been a significant re-evaluation of dietary habits [1,2]. The consumption of nutrient-rich foods or supplements has become very popular among people who strive to attain optimal health benefits [3]. In response to this demand, food industries have developed new healthy products or reformulated existing products. The term “functional food” refers to food that provides more than basic nutrition and has been specially formulated with substances (such as vitamins, minerals, or dietary fiber) or live microorganisms with the potential to enhance health or prevent diseases [4,5]. Several examples include milk fortified with calcium, yogurt enriched with probiotics and prebiotics, omega-3–enriched products, and foods high in antioxidants, including some fruits and vegetables [6]. Currently, probiotics are a hot topic in the field of nutrition, representing an ideal example of a functional food [7,8,9,10].

Probiotics have been utilized for thousands of years, predating the discovery of microorganisms [11]. Their contemporary definition characterizes them as live microorganisms beneficial to host health when administered in adequate quantities [12,13,14,15,16,17]. Most commercially available probiotics strains are *Bifidobacterium*, *Lactobacillus*, and *Saccharomyces boulardii* [18,19]. Each strain of probiotics bacteria makes its own health claim [18]. To have an effective health impact, it is imperative that probiotics ($10^{8}$–$10^{9}$ cfu/g daily) added to food must survive in large numbers throughout the process of fermentation, storage, and consumption [5,20,21,22]. Certain probiotics products are required to be stored in the refrigerator before consumption to keep the probiotics active and to offer health benefits [8,23].

The escalating popularity of probiotics in recent years can be attributed to increased consumer awareness of potential health advantages and a growing interest in natural and alternative wellness approaches [24]. Probiotics have been extensively researched in a number of ways and have demonstrated their benefits and effectiveness in a variety of situations [25,26,27,28,29]. There is evidence that probiotics are beneficial for improving the composition of the gut microbiome, and they can reduce or prevent intestinal inflammation as well as other phenotypes associated with intestinal and systemic diseases [8,30,31,32,33]. There is a growing body of evidence showing that prebiotics and probiotics work synergistically to support a healthy gut environment, resulting in a more effective approach to improving gut health and overall well-being. Prebiotics are defined as non-digestible food constituents, most commonly fiber, which selectively stimulate the growth and/or activity of beneficial microorganisms in the intestines [6]. Keeping the microbiome balanced is crucial to the health of an individual, and the microbiome works in a symbiotic relationship with the host to control nutrient metabolism, protect the host against pathogens, and provide signals to the immune system to enhance host immunity and physiology [34,35,36]. As a result of their numerous health benefits and advertising, the probiotics market is gaining more and more popularity. 

Functional foods with probiotics are increasingly available on the market in many countries. Over the past few years, more than 500 probiotics products have been introduced to the global market, and the number continues to grow [37]. Most commercially available probiotics products are dairy products and fermented products, such as yogurt, cheeses, and sauerkraut [38,39]. Probiotics products include both probiotic foods (such as milk, juice, sourdough bread, and chocolate bars) and dietary supplements (such as tablets, capsules, powders, and liquids) [40,41,42]. Growth of the probiotics market is related to consumer consumption, which is affected by factors such as health consciousness, culture, eating habits, and tastes [43,44]. Aside from these factors, food and beverage manufacturers have embraced the trend by incorporating probiotics into a variety of products, marketing them as promoting digestion and overall well-being. However, food products supplemented with probiotics may not be acceptable to some consumers, and may not be safe or effective for individuals with a variety of medical conditions [45]. Probiotics knowledge enables consumers to make informed decisions about which probiotic foods or supplements are most appropriate for their health or avoid misuse of them. For this reason, understanding the factors that influence consumer acceptance, proper use, as well as the current state of the market are vital for the successful launch of the product and the attainment of health benefits.

It has been reported that probiotics consumption can be affected by factors such as a person’s knowledge, attitude, and education, which have been studied in both the United Arab Emirates [46] and Turkey [40]. Although research on the acceptability of probiotics products has been carried out in different regions, there is a scarcity of studies on this subject in Asia, including Hong Kong. In fact, socio-demographic factors and consumer acceptance levels are regionally dependent, resulting in varying results across the country. Recently, Hong Kong has emerged as one of the most vibrant markets in Asia for health supplements [47]. In 2022, approximately 74% of Hong Kong residents aged 25 to 34 are expected to use nutritional supplements, with roughly 79% taking nutritional supplements in order to boost their immunity, according to the Over the Counter (OTC) & Nutraceuticals Markets Research Report. The city is characterized by a unique blend of Western and Chinese cultures, which influences the dietary practices and health behaviors of its residents [48]. As an international vibrant, cosmopolitan city with a diverse population, Hong Kong offers an intriguing context for studying probiotics acceptance and consumption. 

Based on the findings of previous studies, individuals’ attitudes toward probiotics products and purchase decisions may be influenced by socio-demographic factors (such as age, gender, education, income, and cultural background), health status, and knowledge of probiotics [40,49]. Probiotics products are expanding in the Hong Kong market. However, there is no legislation or standard in Hong Kong governing the dissemination of relevant information regarding the safety and efficacy of dietary supplements. People may be compelled to purchase health-related products as a result of their health-consciousness and business marketing strategy. There is limited research regarding residents’ acceptance of probiotics products and their opinions of these products in Hong Kong. Hence, the purpose of this study is to determine the factors that will influence the decision-making process of Hong Kong residents by systematically assessing socio-demographic factors, health status, and knowledge levels regarding probiotics. Understanding these factors associated with probiotics product acceptability is crucial for business marketing. Additionally, the government can leverage the findings of this study in several ways to inform and enhance public health policies and regulations related to dietary supplements, particularly probiotics products, in Hong Kong. 

## 2. Materials and Methods

### 2.1. Experimental Design

A comprehensive quantitative and descriptive investigation was undertaken utilizing a meticulously crafted questionnaire to analyze consumers’ reception and perspectives regarding probiotics consumption. The questionnaire comprised 27 questions segmented into four sections, each presenting multiple choices for participants to select from. Employing a non-random sampling technique, the survey was administered from June through mid-July 2024, with interviews conducted via online platforms, telephone, and face-to-face interactions. Participants completed the questionnaires within a time frame of 15 to 20 min. The sampling phase extended over approximately one and a half months. Before conducting the research, ethical approval by the Technological and Higher Education Institute of Hong Kong’s Human Subjects Ethics Committee (Ref. No.: SHE2024-008) was obtained on 5 March 2024.

### 2.2. Subject Recruitment and Sample Size

The study aimed to provide a comprehensive assessment of the socio-demographic factors, health status, and knowledge regarding probiotics product acceptability in Hong Kong. The target participants were Hong Kong residents and those who had resided in Hong Kong for a period of more than 6 months. To ensure that the data collected is exclusively from the Hong Kong market, the first two questions of the questionnaire asked about their willingness to participate and their residence.

As the sample size of the survey has a direct bearing on the quality and reliability of the results, it should be carefully selected to reflect the target population. Sample size for the current study was calculated based on a previous study, which determined sample size by using the standard error formula (Equation (1)) [40].
(1)$$n=\frac{X^{2}\cdot NP(1-P)}{d^{2}\left(N-1\right)+X^{2}P(1-P)}$$

n is the estimated sample size, X is the confidence interval (95% confidence is = 1.96), N is the size of the population, P is the sample portion (50%), and d is the degree of accuracy (0.05). In accordance with the 2021 census, Hong Kong has an estimated population of 7.41 million. The optimal sample size was calculated as: $\left[1.96^{2}\cdot 7410000\left(0.5\right)\left(1-0.5\right)\right]/[0.05^{2}\left(7410000-1\right)+1.96^{2}0.5\left(1-0.5\right)]\;\approx 384$. Thus, a total of 385 participants were recruited for this study.

### 2.3. Socio-Demographic Characteristics of the Respondents

A frequency analysis was conducted to describe the socio-demographic characteristics of the 385 participants in the study (Table 1). Among the participants, the number of female participants (51.9%) is slightly higher than the male participants (47.8%), and those between the ages of 46 and 65 constituted the largest group (27.5%). As a means of obtaining a more accurate picture of the Hong Kong market, the participants’ age distribution was comparable to that of the Hong Kong population. Based on a census conducted by the Hong Kong Census and Statistics Department in June and August 2011, approximately 16% of Hong Kong’s population was under the age of 18, 6% were between the ages of 18 and 25, 13% were between the ages of 26 and 35, 14% were between the ages of 36 and 45, 31% were between the ages of 46 and 65, and 20% were over the age of 66.

### 2.4. Data Analysis

This study collected data using an online questionnaire developed from a Google form that consisted of multiple-choice questions. Tables and graphs were prepared using IBM SPSS Statistics version 28 for Windows and Microsoft Office Excel 365. The questionnaire’s reliability and internal consistency were assessed using Cronbach’s alpha reliability at a significance level of 0.05. Data normality was examined through both Kolmogorov–Smirnov and Shapiro–Wilk normality tests at the 0.05 significance level. The difference between scores was checked statistically using the Chi-squared test. The interaction between study variables was determined using Pearson correlation and regression analysis at the 0.05 level, aiming to ascertain the presence of a statistically significant relationship among the variables.

## 3. Results

### 3.1. Respondents’ Socio-Demographic Profile

In this study, 15.1% of respondents were under the age of 18, 16.4% were between 18 and 25 years old, 16.1% were between 26 and 35 years old, 10.6% were between 36 and 45 years old, 27.5% were between 46 and 65 years old, and approximately 14.3% were over 66 years old. The majority of respondents were over the age of 18 and were predominantly married (56.1%). A majority of respondents (28.6%) completed secondary education, followed by those with a bachelor’s degree (23.9%), master’s degrees or higher (21.6%), college (14.8%), and primary education (11.2%). The data revealed that most participants had a degree of education. The proportion of students responding to the questionnaire was the highest (22.9%), followed by professionals (12.2%). Among the respondents, those with incomes between HK $20,000 and 39,999 (37.7%) accounted for the majority, followed by those with incomes between HK $40,000 and above (25.5%) and those with incomes between HK $10,000 and 19,999 (24.2%).

It is notable that women had higher education, with 15.32% having obtained a bachelor’s degree and 11.95% having obtained a master’s degree or above, respectively, while 8.57% of men had bachelor’s degrees and 9.35% had master’s degrees (Figure 1). But more male respondents had primary or below (7.79%) and college level (8.83%) than females.

### 3.2. Health Assessments of Respondents

As part of the questionnaire, five questions were designed to obtain information regarding respondents’ health status. Among respondents who were asked to indicate their weight and exercise frequency, the majority perceived themselves to be of normal weight (70.6%) and most exercised regularly, but not more than 150 min per week of moderate-intensity aerobic exercise (35.5%) (Table 2). In order to assess respondents’ perceptions of their weight and height and the true prevalence, their body mass index (BMI) was calculated using the information they provided about their height and weight. It was found that the results slightly differed from what they reported. A majority of respondents were within the normal weight range, but a greater number of participants were actually underweighted (19.8%) than they reported (9.6%).

In response to the question ‘Do you have any of these health problems (you may answer more than one question)’, the majority of respondents reported no illness (57.9%), followed by gastrointestinal disease (15.3%) and allergies (10.4%) (Table 3). Females generally suffered from fewer health issues than males when gender and health were considered together. A higher percentage of females reported no disease, and there was a greater prevalence of obesity (71.4%), diabetes (63.3%), hypertension (72.4%), coronary heart disease (100%), and high cholesterol (71.4%) among males. The most common health problems reported by females were gastro-intestinal (72.4%), followed by osteoporosis (64.3%) and allergy (61.5%).

### 3.3. Respondents’ Awareness and Knowledge of Probiotics

As part of the questionnaire, nine yes/no questions were included to assess respondents’ knowledge of probiotics (Table 4). The first three questions assessed respondents’ understanding of the definition of probiotics, the next three questions addressed the potential negative effects of probiotics, and the final three questions examined the mechanism of action of probiotics. There was a total score of nine points and an average score of five points among the 385 respondents (Figure 2). In accordance with the results of the test, most respondents had a fair understanding of probiotics (Table 4). It was found that 77.9% of respondents had a basic knowledge of probiotics, while approximately half were unaware of the concept of prebiotics (44.2%). Most respondents were aware of the relationship between probiotics and gut function (90.4%) and believed that probiotics were safe for everyone (69.9%).

### 3.4. Correlation between Socio-Demographic Factors, Health Status, and Knowledge

There was an examination of the Pearson correlation coefficients (r) and asymptotic significance (2-tailed) (*p*-value) for 10 variables pertaining to socio-demographic characteristics, health status, and knowledge on probiotics among 385 respondents (Table 5). The Pearson correlation coefficient (r) is a statistical test statistic used to measure a statistical relationship between two continuous variables. A greater r coefficient indicates a stronger correlation between the variables, and positive Pearson correlation coefficients indicate an increase in one value is accompanied by an increase in the other value. *p*-values are used in hypothesis testing to assist with the decision to accept or reject the null hypothesis. If the *p*-value is less than 0.05, it is more likely that the null hypothesis will be rejected.

Data indicates that some socio-demographic variables are strongly positively correlated. It was found that the strongest association was between ‘Education Attainment’ and ‘Monthly household income’ (r = 0.273, *p* < 0.001), suggesting that the more educated the respondents, the higher their monthly income. Besides, there was a strong positive correlation between ‘Age’ and ‘Monthly household income’ (r = 0.269, *p* < 0.001), ‘Educational Attainment’ and ‘Score’ (r = 0.257, *p* < 0.001), ‘Marital Status’ and ‘Risk Perception’ (r = 0.216, *p* < 0.001), and ‘Monthly household income’ and ‘Score’ (r = 0.232, *p* < 0.001). The *p*-value for these associations was less than 0.05, indicating that the correlation was statistically significant and that the two variables were related. ‘Score’ refers to the results of the probiotics knowledge test in the questionnaire among respondents. Accordingly, probiotics knowledge was closely related to socio-demographic characteristics, and respondents with a higher level of education and monthly income had a higher knowledge of probiotics. Additionally, the results noted that risk awareness of probiotics products was associated with marital status. The positive correlation (r = 0.216, *p* < 0.001) refers to probiotics products that were perceived to be riskier by married participants (Figure 3). In terms of the negative relationship, there was a negative correlation between ‘Monthly household income’ and ‘Consumption frequency’ (r = −0.121, *p* = 0.018), that is, the lower the income, the lower the probability of probiotics consumption.

Considering the correlation between respondents’ health status and variables (Table 6), age is closely related to health problems. With the exception of ‘Allergy’ and ‘No disease’, all other health problems were strongly positively correlated with age, especially the associations between ‘Age’ and ‘Diabetes’ (r = 0.258, *p* < 0.001), ‘Age’ and ‘Hypertension’ (r = 0.282, *p* < 0.001), ‘Age’ and ‘Cholesterol’ (r = 0.207, *p* < 0.001), and ‘Age’ and ‘Osteoporosis’ (r = 0.203, *p* < 0.001). The strongest association was between ‘Weight Define’ and ‘Obesity’ (r = 0.340, *p* < 0.001), suggesting that respondents who reported they were overweight were more likely to be obese and that the self-report was accurate.

### 3.5. Consumption Habits and Risk Awareness of Probiotics Products

Nearly all respondents (94.8%) believed that probiotics products were readily available in Hong Kong (Table 7). Most respondents consumed and took probiotics products on a regular basis. The majority of respondents consumed probiotics products two times a week (19.5%) or three times a week or more (19.5%) (Table 8). However, the fewest respondents consumed probiotics products three times a day or more (0.8%). There were 11.2% of respondents who reported that they had never consumed probiotics products before. Most of them consumed probiotics irregularly (48.1%), and they are equally likely to take them after meals (17.7%) or with meals (16.1%) (Table 7). It appears that most respondents believed that probiotics posed some risks (61.6%), primarily due to ‘Improper production process’ (13%), ‘Low-quality probiotics product’ (20.3%), ‘Potential negative effects on the body’ (21%), and ‘Overstating benefits’ (22.9%) (Table 7). Among the respondents, fermented foods were their primary source of probiotics (60.5%), and dietary supplements (31.9%) were preferred over homemade probiotics products (6%). Besides, it was found that 37.9% of women chose probiotics products because of their health benefits, and 13.1% said it was due to advertising or marketing efforts, both of which were higher than for men (Table 9).

## 4. Discussion

Consumer acceptance of probiotics products is a one of the key determinants of purchasing behavior. Probiotics have been found to have multiple health benefits for humans and can be used as a therapeutic treatment in certain circumstances [5,50,51,52]. It is generally considered safe to consume probiotics [53,54]. Promoting the consumption of probiotics products may be an effective strategy for improving people’s health [55,56]. Thus, the purpose of this study was to examine the acceptability of probiotics products in the Hong Kong market through an assessment of consumer-related factors, including socio-demographic factors, health status, and probiotics knowledge, among Hong Kong residents. The results for these variables are presented in the Results section and will be discussed further below.

According to this study, a greater percentage of women had higher educational attainment than men (Figure 1), and there was a positive correlation between ‘Educational Attainment’ and ‘Score’ (r = 0.257, *p* < 0.001) (Table 5). It means that respondents with higher scores on the probiotics knowledge test tend to have higher levels of education. There is evidence to suggest that women with higher education backgrounds possess a greater knowledge of probiotics than men. In this regard, the improvement of education may contribute to an increased level of health-related knowledge, which will subsequently result in a positive impact on health outcomes. A similar study was conducted in Saudi Arabia examining socio-demographic differences in probiotics knowledge, which found that males and females had significantly different probiotics knowledge scores at the same education level, but males scored much higher on knowledge tests than females [57]. This study was conducted among health science students and revealed that the age group of 25 to 26 years old performed better on the knowledge test than the other groups [57]. They set narrower age groups, which included 18–20, 21–22, 23–24, and 25–26. However, in the present study, the age group setting was wide, and there was no significant interaction between age and knowledge test. There may be a relationship between age and probiotics knowledge within a more specific age range. Therefore, in the next study, it is advisable to narrow the age range to study the relation between age and probiotics knowledge. A further limitation of the present study is that it does not consider the educational background of the respondents. People with a science background are more familiar with the concept of probiotics.

In addition to finding that female respondents possess a higher level of education and score higher on the probiotics knowledge test, the questionnaire found that more female respondents had not been diagnosed with any disease, and fewer reported chronic diseases (e.g., Diabetes, Hypertension, and Coronary heart disease) (Table 3). According to the latest data from the Centre for Health Protection of the Department of Health, the life expectancy of women in Hong Kong is 88 years, whereas that of men is 82 years. The mortality rate for women around the world was also lower [58]. Additionally, the study found that 37.9% of women chose probiotics primarily because of their health benefits and 13.1% because of advertising or marketing efforts, both of which were higher than the rate for men. The education level is likely to be a contributing factor to a greater number of females being healthy. Women are likely to be the first target of supplement manufacturers since they tend to purchase supplement products and pay attention to product claims regarding their health. Typically, women make the majority of decisions about family food choices, and a healthy diet can aid in preventing and controlling diseases, so the health status of family members may be directly related to women’s health consciousness. It is therefore extremely important to educate women on correct health knowledge so that misleading information will not be spread. A health improvement strategy that involves educating men about health knowledge as well as encouraging them to pursue higher education may be effective. While there was no significant difference in the consumption of probiotics between genders, females tend to consume more probiotics products (once a day, *n* = 32) (Table 8), indicating a slightly higher acceptance of these products among women.

In addition to gender differences in probiotics awareness, income may be a factor affecting probiotics purchases. Considering the market price for novel foods is generally higher than the market price for traditional foods [40], family income may be related to the consumption of probiotics products. Research in the past has shown that people with higher household incomes tend to consume higher-quality diets, prefer healthy foods, and have more disposable income to spend on healthy items [59]. In this study, 7.8% of respondents reported incomes below HK $10,000, 24.2% reported incomes between HK $10,000 and $19,999, 37.3% reported incomes between HK $20,000 and $ 39,999, 25.5% reported incomes over HK $ 40,000, and 4.9% reported that they were either unemployed or retired. A report from the Consumer Council of Hong Kong examined 40 probiotic dietary supplements on the market, with prices ranging from $99 to $788. The price of the probiotics product does not appear to be a significant factor that will deter Hong Kong consumers from purchasing it.

The results reveal that respondents possess a reasonable comprehension of probiotics (Figure 2). In this study, a segment of the questionnaire comprised nine yes/no questions to evaluate respondents’ knowledge of probiotics (Table 4). The responses to the initial three questions indicate that the majority of participants can recognize probiotics as non-drug substances and acknowledge their role in enhancing gut health by rebalancing the gut microbiota. But prebiotics were not clearly defined by approximately half of the respondents. It is suggested that the basic concept of probiotics is likely to be familiar to most Hong Kong residents. This may be due to the fact that probiotics have been mentioned more frequently than prebiotics in the market. Using the term ‘probiotics’ can find several related articles, while using the term ‘prebiotics’ cannot seem to find any related articles, according to the research results of the Consumer Council in Hong Kong. The information regarding probiotics attracts more attention. Recently, the Consumer Council conducted a market survey based on overseas standards and reviewed 40 probiotics products but has not yet researched prebiotics products. The majority of respondents agreed that probiotics are safe (69.9%). Nearly half of the respondents believe that probiotics may be used as drugs to treat diseases. It appears most people may be aware of the connection between probiotics and gut health through advertising and education, but may still be uncertain about the actual role and impact of probiotics, which may lead to inappropriate use. Thus, it is imperative to inform individuals about the precautions when using probiotics. In reviewing the findings of the last three questions about probiotics mechanisms of action, it appears that more than half of the respondents were aware that probiotics produce positive effects through their action in the gut. However, most were unaware of the interaction between probiotics and other organs or tissues. Several respondents (54.4%) expressed disagreement with the idea that probiotics can produce neurotransmitters and other signaling molecules that influence mood and cognitive function. In fact, probiotics are capable of doing more than improving intestinal health; they may also have an indirect effect on the brain via the Gut–Brain Axis [60]. A study found that probiotics could improve mood and reduce depressive disorders through the alteration of gut microbiota [61]. Consequently, it is suggested that there is a connection between the brain and the gut, and that gut health is of great importance. According to these findings, respondents were familiar with basic concepts of probiotics but did not fully comprehend their potential roles and effects on the health of humans. For the purpose of preventing the improper use of probiotics by consumers as well as the importance of gut function to human health, both the government and manufacturers should be responsible for tightening surveillance and marketing the products properly.

Considering the limitations of this study and points to be considered in future investigation, the total number of participants in the survey (385) may not be representative of Hong Kong residents compared to the more than 2000 participants in the government survey. To obtain more representative data for further study, it is recommended to launch a study with a larger number of participants to examine the acceptability of probiotics products in Hong Kong. The age group set in the questionnaire was too broad and does not specifically include younger children (under 10 years old) or older populations (over 80 years old). People from diverse backgrounds are encouraged to participate in further studies. Furthermore, the questionnaire did not include questions regarding mental health and factors that may affect consumption, nor did it directly ask participants whether they were willing to purchase probiotics. Therefore, incorporating these factors into future studies will result in more accurate results.

## 5. Conclusions

There is growing popularity of probiotics products in the food market through innovative marketing methods, and researchers are paying increasing attention to the subject as well. Consumers’ demand and acceptance play a crucial role in enticing manufacturers to develop such novel products to improve health, well-being, and quality of life. As a result of the findings, it is suggested that probiotics products in Hong Kong are generally well-accepted and well-distributed. Probiotics product acceptance differs by gender and is influenced by education level. Females with higher levels of education could be the primary target of probiotics products. Most Hong Kong residents have a basic understanding of probiotics, but they are not fully acquainted with them and are uncertain about their proper use. As Hong Kong currently does not have any specific legislation or standards governing the provision of information regarding relevant safety and efficacy of dietary supplements, Hong Kong residents are advised to know the correct information and how to use probiotics properly by themselves to prevent side effects of misuse. According to the study results, the majority of Hong Kong residents can afford the price of novel probiotics products. With the increasing health awareness, it is expected that more and more people will buy probiotics products in the future, and the quality and safety of probiotics products are the major concerns of consumers. A report published by the Consumer Council in 2024 concluded that probiotics products on the market are chaotic and unreliable. *Enterococcus faecalis* was detected in two samples, which is an unstable strain, easily contaminated, and not recommended for use as a probiotic in humans. Moreover, most probiotics products have incomplete labeling and lack sufficient scientific evidence to support certain claims of efficacy. This study indicates a relationship between probiotics understanding and education attainment. The positive correlation suggests that encouraging individuals to pursue higher education may be an effective strategy for improving health outcomes. Considering the results of this study, people should be motivated to read more about scientific research relevant to their lives in order to obtain unbiased health information. And the Hong Kong government has the responsibility to pay more attention to this and urgently introduce relevant policies. As well, schools are encouraged to incorporate extended education into all major curricula in order to enhance students’ understanding of the current topic. Incorporate educational programs that provide information on probiotics and prebiotics, identify probiotic foods and probiotic supplements, and provide considerations when consuming probiotics to improve health outcomes.

Studying the consumer’s acceptance of novel products can provide valuable information to manufacturers in designing marketing strategies. It may also serve as the basis for developing regulations by government agencies. In this study, only a limited number of consumer characteristics, preferences, health status, and probiotics knowledge were considered to assess the acceptability of probiotics products. To gain a more comprehensive understanding of consumer acceptance of probiotics products, more research is needed, including the relationship between education background and probiotics knowledge, the effect of cultural background on probiotic foods selection, and the impact of income on the acceptance of probiotics products.

## Figures and Tables

**Figure 1 foods-13-02971-f001:**
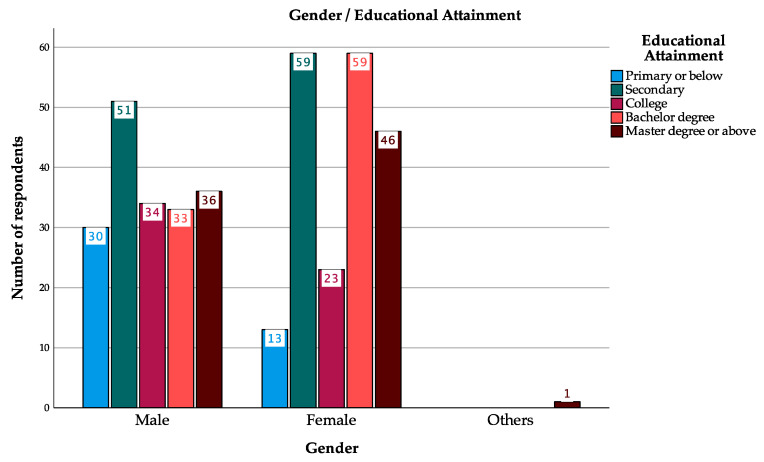
Gender and Educational Attainment.

**Figure 2 foods-13-02971-f002:**
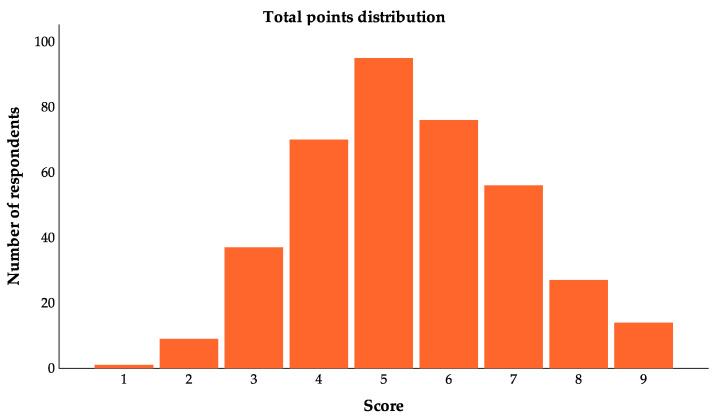
Points distribution among the probiotics knowledge test.

**Figure 3 foods-13-02971-f003:**
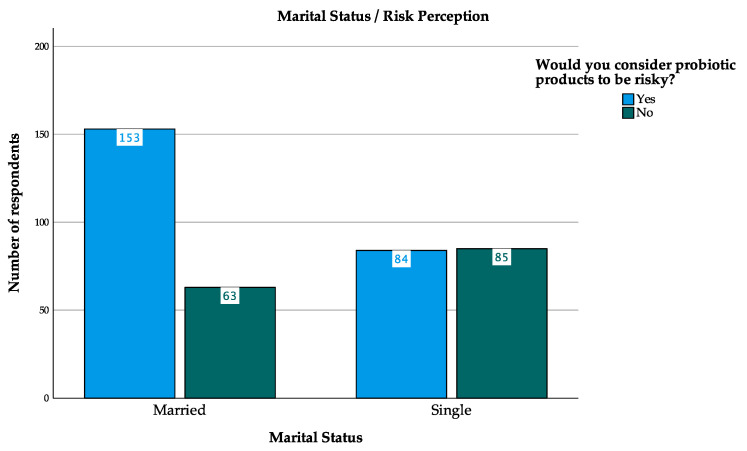
Marital Status and Risk Perception.

**Table 1 foods-13-02971-t001:** Socio-demographic characteristics of the respondents.

Variables	Response	Frequency (*n*)	Percentage (%)
Gender	Male	184	47.8
	Female	200	51.9
	Others	1	0.3
Age Group	Below 18 years old	58	15.1
	18–25 years old 26–35 years old 36–45 years old 46–65 years old 66 years old and older	63 62 41 106 55	16.4 16.1 10.6 27.5 14.3
Marital Status	Married	216	56.1
	Single	169	43.9
Educational attainment	Primary and below Secondary College Bachelor’s degree Master’s degree or above	43 110 57 92 83	11.2 28.6 14.8 23.9 21.6
Occupation	Managers and administrators Professionals Associate professionals Clerical support workers Service and sales workers Agricultural and fisheries Craft and related workers Drivers and machine operators Technicians Unskilled workers Students Unemployed or retired	28 47 26 35 45 7 6 12 27 36 88 21	7.3 12.2 6.8 9.1 11.7 1.8 1.6 3.1 7.0 9.4 22.9 5.5
Monthly household income	Less than HK $10,000 HK $10,000–$19,999 HK $20,000–$39,999 HK $40,000 and above Unemployed or retired	30 93 145 98 19	7.8 24.2 37.7 25.5 4.9

**Table 2 foods-13-02971-t002:** Participants’ weight and exercise status.

Variables	Responses	Frequency (*n*)	Percentage (%)
How do you define your weight?	Underweight	37	9.6
	Normal weight	272	70.6
	Overweight	76	19.7
Physical performance (based on BMI)	Underweight	76	19.8
	Normal weight	238	62.0
	Overweight	62	16.1
	Obesity	8	2.1
Exercise frequency	At least 150 min of moderate-intensity aerobic exercise per week	129	33.5
	Less than 150 min of moderate-intensity aerobic exercise per week	136	35.3
	Rarely exercise	120	31.2

**Table 3 foods-13-02971-t003:** Health status of the respondents.

Health Problems	Responses		Percent of Cases	Gender	
	N	Percent		Male (%)	Female (%)
Obesity	15	3.4%	3.9%	71.4%	28.6%
Diabetes	31	7.0%	8.1%	63.3%	36.7%
Hypertension	30	6.8%	7.8%	72.4%	27.6%
Coronary heart disease	7	1.6%	1.8%	100.0%	0.0%
Cholesterol	22	5.0%	5.7%	71.4%	28.6%
Osteoporosis	15	3.4%	3.9%	35.7%	64.3%
Allergy	40	9.0%	10.4%	38.5%	61.5%
Gastro-intestinal	59	13.3%	15.3%	27.6%	72.4%
No disease	223	50.5%	57.9%	46.6%	53.4%

**N**: The number of people who selected each option. **Percent**: The ratio between the number of selections for each option and the total number of selections. **Percent of Cases**: The percentage of people who choose this item out of the total number of people.

**Table 4 foods-13-02971-t004:** Participants’ responses to the probiotics knowledge test.

Question	Response	
	Correct Answer (%)	Incorrect Answer (%)
Do you think the essence of probiotics is drugs?	77.9	22.1
Do you consider prebiotics to be a food source?	55.8	44.2
Do you think probiotics can help restore the balance of gut microbiota and promote the growth of beneficial bacteria?	90.4	9.6
Do you think consuming probiotics is safe for everyone, with few reported side effects?	30.1	69.9
Do you think probiotics can be used as drugs to treat certain health conditions (such as irritable bowel syndrome, inflammatory bowel disease)?	50.9	49.1
Do you think probiotics will react with certain drugs and cause adverse effects?	60.8	39.2
Do you think probiotics work by directly colonizing the gut and displacing harmful bacteria with beneficial ones?	67.3	32.7
Do you think the short-chain fatty acids produced by probiotics can inhibit the growth of harmful bacteria in the gut?	59.2	40.8
Do you think probiotics can produce neurotransmitters and other signaling molecules that can affect mood and brain function?	45.5	54.4

**Table 5 foods-13-02971-t005:** Pearson correlation coefficients and Asymptotic significance among 10 variables.

		Gender	Age	Marital Status	Educational Attainment	Occupation	Monthly Income	Weight Define	Score	Consumption Frequency	Risk Perception
Gender	r	1	−0.062	0.066	0.143 **	−0.076	0.015	0.005	0.089	−0.107 *	0.057
	*p*-value		0.225	0.199	0.005	0.135	0.769	0.919	0.080	0.036	0.268
Age	r	−0.062	1	−0.759 **	−0.118*	−0.252**	0.269**	−0.098	0.042	−0.074	−0.167 **
	*p*-value	0.225		<0.001	0.021	<0.001	<0.001	0.054	0.409	0.145	0.001
Marital Status	r	0.066	−0.759 **	1	0.155 **	0.190 **	−0.254 **	0.068	−0.048	0.070	0.216 **
	*p*-value	0.199	<0.001		0.002	<0.001	<0.001	0.185	0.351	0.171	<0.001
Educational Attainment	r	0.143 **	−0.118 *	0.155 **	1	−0.485 **	0.273 **	0.003	0.257 **	−0.051	0.028
	*p*-value	0.005	0.021	0.002		<0.001	<0.001	0.959	<0.001	0.319	0.578
Occupation	r	−0.076	−0.252 **	0.190 **	−0.485 **	1	−0.257 **	−0.022	−0.210 **	0.116 *	−0.006
	*p*-value	0.135	<0.001	<0.001	<0.001		<0.001	0.673	<0.001	0.023	0.910
Monthly Income	r	0.015	0.269 **	−0.254 **	0.273 **	−0.257 **	1	−0.006	0.232 **	−0.121 *	−0.061
	*p*-value	0.769	<0.001	<0.001	<0.001	<0.001		0.903	<0.001	0.018	0.232
Weight Define	r	0.005	−0.098	0.068	0.003	−0.022	−0.006	1	−0.057	−0.045	0.090
	*p*-value	0.919	0.054	0.185	0.959	0.673	0.903		0.261	0.380	0.077
Score	r	0.089	0.042	−0.048	0.257 **	−0.210 **	0.232 **	−0.057	1	−0.164 **	−0.163 **
	*p*-value	0.080	0.409	0.351	<0.001	<0.001	<0.001	0.261		0.001	0.001
Consumption Frequency	r	−0.107 *	−0.074	0.070	−0.051	0.116 *	−0.121 *	−0.045	−0.164 **	1	−0.094
	*p*-value	0.036	0.145	0.171	0.319	0.023	0.018	0.380	0.001		0.067
Risk Perception	r	0.057	−0.167 **	0.216 **	0.028	−0.006	−0.061	0.090	−0.163 **	−0.094	1
	*p*-value	0.268	0.001	<0.001	0.578	0.910	0.232	0.077	0.001	0.067	0.013

r: Pearson correlation; *p*-value: Asymptotic significance (2-tailed). ** Correlation is significant at the 0.01 level (2-tailed). * Correlation is significant at the 0.05 level (2-tailed).

**Table 6 foods-13-02971-t006:** Correlations between health problems and variables.

		Gender	Age	Marital Status	Educational Attainment	Monthly Income	Weight Define	Exercise Frequency	Consumption Frequency	Risk Perception
Obesity	r	−0.050	0.013	0.038	0.026	0.022	0.340 **	0.039	−0.030	0.089
	*p*-value	0.0330	0.794	0.454	0.614	0.663	<0.001	0.443	0.561	0.080
Diabetes	r	−0.062	0.258 **	−0.166 **	−0.135 **	−0.025	0.231 **	−0.075	0.055	−0.057
	*p*-value	0.227	<0.001	0.001	0.008	0.624	<0.001	0.144	0.283	0.263
Hypertension	r	−0.110 *	0.282 **	−0.238 **	−0.107*	0.081	0.163 **	−0.148 **	−0.043	−0.090
	*p*-value	0.031	<0.001	<0.001	0.036	0.115	0.001	0.004	0.402	0.077
Coronary heart disease	r	−0.064	0.122 *	−0.081	−0.074	0.006	0.011	−0.044	−0.020	−0.068
	*p*-value	0.207	0.016	0.112	0.147	0.907	0.835	0.385	0.695	0.186
Cholesterol	r	−0.079	0.207 **	−0.128 *	−0.029	0.100 *	0.163 **	−0.090	0.036	−0.080
	*p*-value	0.124	<0.001	0.012	0.564	0.049	0.001	0.077	0.483	0.119
Osteoporosis	r	0.083	0.203 **	−0.124 *	−0.164 **	−0.005	−0.114 *	0.089	−0.030	−0.104 *
	*p*-value	0.103	<0.001	0.015	0.001	0.929	0.025	0.080	0.561	0.042
Allergy	r	0.085	−0.095	0.110 *	0.092	0.066	−0.001	0.137 **	−0.003	0.028
	*p*-value	0.098	0.064	0.030	0.071	0.196	0.987	0.007	0.959	0.578
Gastro-intestinal	r	0.186 **	0.014	0.031	0.013	−0.032	−0.054	0.111 *	−0.019	0.064
	*p*-value	<0.001	0.779	0.550	0.793	0.536	0.292	0.029	0.706	0.210
No disease	r	0.021	−0.309 **	0.245 **	0.153 **	−0.064	−0.095	−0.110 *	−0.046	0.003
	*p*-value	0.684	<0.001	<0.001	0.003	0.211	0.063	0.031	0.365	0.954

r: Pearson correlation; *p*-value: Asymptotic significance (2-tailed). ** Correlation is significant at the 0.01 level (2-tailed). * Correlation is significant at the 0.05 level (2-tailed)

**Table 7 foods-13-02971-t007:** Participants’ responses to probiotics consumption habits and risk awareness.

Variables	Response	Frequency (*n*)	Percentage (%)
Do you think probiotics products are readily available?	Yes	365	94.8
	No	20	5.2
Would you consider probiotics products to be risky?	Yes	237	61.6
	No	148	38.4
What is your biggest concern about probiotics products?	Improper production process	50	13.0
	Low-quality probiotics products	78	20.3
	Potential negative effects on the body	81	21.0
	Overstating benefits	88	22.9
	No concerns	88	22.9
How do you take probiotics?	Fermented food (e.g., yogurt, cheese)	233	60.5
	Dietary supplements	123	31.9
	Homemade probiotics products	23	6.0
When do you take probiotics?	On an empty stomach	32	8.3
	Take before meals	28	7.3
	Take after meals	68	17.7
	Take with meal	62	16.1
	Irregular intake	185	48.1

**Table 8 foods-13-02971-t008:** Probiotics products consumption among respondents.

Variables	Response	Frequency (*n*)	Percentage (%)	Gender	
				Male (*n*)	Female (*n*)
How often do you consume probiotics products?	Once a day	50	13.0	18	32
	Two times a day	13	3.4	4	9
	Three times a day or more	3	0.8	1	1
	Once a week	47	12.2	24	23
	Two times a week	75	19.5	38	37
	Three times a week or more	75	19.5	35	40
	Every 15 days	48	12.5	25	23
	Never tried	43	11.2	22	21
	Irregular frequency	31	8.1	17	14

**Table 9 foods-13-02971-t009:** Gender and Reasons for choosing probiotics products.

Gender	Reasons					
		Advertisement/Promotion	Recommendation	Diet/Lifestyle	Health Benefits	Taste
Male	Count	28	28	72	85	63
	% within Gender	10.1%	10.1%	26.1%	30.8%	22.8%
Female	Count	44	35	76	127	127
	% within Gender	13.1%	10.4%	22.7%	37.9%	15.8%

## Data Availability

The original contributions presented in the study are included in the article, further inquiries can be directed to the corresponding author.

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
