# Peer review of "How Do Socio-Demographic Factors, Health Status, and Knowledge Influence the Acceptability of Probiotics Products in Hong Kong?"

_foods, 2024, doi:10.3390/foods13182971_

Round 1

Reviewer 1 Report

Comments and Suggestions for Authors

Thank you for the opportunity to review the manuscript titled "How do sociodemographic factors, health status, and knowledge influence the acceptability of probiotic products in Hong Kong?" Although the topic may not be particularly novel or original, it could still be relevant to the readers of Foods if the authors better focus the objective within a more clearly defined conceptual framework. Currently, the manuscript feels fragmented and lacks clarity.

I believe the literature review and discussion sections are the areas most in need of improvement, starting with a more defined and focused conceptual background. To enhance the paper, I recommend that the authors identify more targeted research questions grounded in a solid theoretical framework, and expand the literature review section by including additional references and engaging more deeply with the findings.

In addition, systematic reviews on functional foods and probiotics could further assist the authors in strengthening this section.

Methodology

Section 3.1, which discusses the socio-demographic characteristics of the respondents, should be moved to the methodology section. This information pertains to the sample's socio-demographic characteristics and is not a result of the study.

Results

The results are well-written and clearly described.

Discussion

It would be beneficial to include a more comprehensive discussion of previous similar results from other countries and other emerging economies in the discussion section. This would provide a broader context for the study's findings and help readers understand how the results can be compared to existing research with similar topic in other countries.

Pag 13 Line 345-250: This part is not consistent with the discussion and should either be moved to another section of the paper or deleted. “ The Census and Statistics Department ………………………………affordable for most Hong Kong residents.

Conclusion

This section is also incomplete. The authors should discuss the main limitations of their study and suggest potential avenues for future research. Additionally, they should highlight the key implications for marketing, theoretical development, and policy-making.

Author Response

Thank you for the opportunity to review the manuscript titled "How do sociodemographic factors, health status, and knowledge influence the acceptability of probiotic products in Hong Kong?" Although the topic may not be particularly novel or original, it could still be relevant to the readers of Foods if the authors better focus the objective within a more clearly defined conceptual framework. Currently, the manuscript feels fragmented and lacks clarity.

I believe the literature review and discussion sections are the areas most in need of improvement, starting with a more defined and focused conceptual background. To enhance the paper, I recommend that the authors identify more targeted research questions grounded in a solid theoretical framework, and expand the literature review section by including additional references and engaging more deeply with the findings.

In addition, systematic reviews on functional foods and probiotics could further assist the authors in strengthening this section.

Response: Thank you for your insightful feedback. We have taken your suggestions into careful consideration and have made significant revisions to the sections “1. Introduction” and “4. Discussion” of the manuscript. The updated sections now provide a more coherent and structured narrative that enhances the overall flow and understanding of the study. Additionally, we have expanded the literature review by incorporating additional references (please refer to the highlighted parts in both sections).

Methodology

Section 3.1, which discusses the socio-demographic characteristics of the respondents, should be moved to the methodology section. This information pertains to the sample's socio-demographic characteristics and is not a result of the study.

Response: Section 3.1, which discusses the socio-demographic characteristics of respondents, has been moved to the methodology session.

Results

The results are well-written and clearly described.

Response: Thank you!

Discussion

It would be beneficial to include a more comprehensive discussion of previous similar results from other countries and other emerging economies in the discussion section. This would provide a broader context for the study's findings and help readers understand how the results can be compared to existing research with similar topic in other countries.

 Response: The literature review and discussion section now include a comparison with similar results from other countries

Pag 13 Line 345-250: This part is not consistent with the discussion and should either be moved to another section of the paper or deleted. “ The Census and Statistics Department ………………………………affordable for most Hong Kong residents.

Response: The sentence "Hong Kong households' median monthly income in 2022 was HK $28,300" has been removed from the discussion.

Conclusion

This section is also incomplete. The authors should discuss the main limitations of their study and suggest potential avenues for future research. Additionally, they should highlight the key implications for marketing, theoretical development, and policy-making.

Response: The conclusion has been revised based on the suggestions (please refer to section  “5. Conclusion”).  

Reviewer 2 Report

Comments and Suggestions for Authors

The manuscript is of interest for researchers. However, I would like to point out some suggestions that could be addressed. 

1. Please number the equations and address them in text. 

2. Please explain better the meaning of N and P in equation 1.

3. Please explain if you asked for a consent form. Is there an ethical approval from your research center? Was the Helsinki protocol followed? How were the underaged participants addressed?

4. Please explain a little bit further, how was the questionnaire applied to subjects

5. One of the points that you established in your conclusions are: " This study indicates a relationship between probiotic understanding 410 and education attainment. The positive correlation suggests that encouraging individuals 411 to pursue higher education may be an effective strategy for improving health outcomes. 412 Educating the public about probiotics and prebiotics, distinguishing probiotics foods and 413 probiotics supplements, and considerations when consuming probiotic products ". 

In the introduction and in the discussions this matter is not addressed,  therefore it is important to establish what has been done up to now regarding education in probiotics.

What could we do with the  information of this project? 

On the other hand, the research focuses in subjects with different ages and education backgrounds.  Based on that, is it possible to establish if  people have more knowledge in probiotics based on their age.  Or is there a relation between their education background and the knowledge of probiotics?

What are the main topics that should be learned on probiotics by the people in this country. 

Consider the above to evaluate your conclusions.

In addition, add in conclusions what are the next steps for this research. How will this knowledge contribute to society.  And establish what are the limits of your study.

Author Response

The manuscript is of interest for researchers. However, I would like to point out some suggestions that could be addressed. 

  1. Please number the equations and address them in text. 

Response: The equation has been numbered and addressed in the text (please refer to line: 141).

  1. Please explain better the meaning of N and P in equation 1.

Response: The meaning of all unknowns has been added (please refer to section “2.2. Subject Recruitment and Sample Size”).

  1. Please explain if you asked for a consent form. Is there an ethical approval from your research center? Was the Helsinki protocol followed? How were the underaged participants addressed?

Response: The information about the ethical approval has been added to section “2.1 Experimental Design” (please refer to line 126-128). The underaged participants will be analyzed under the group of 18 years old and younger.

  1. Please explain a little bit further, how was the questionnaire applied to subjects

Response: We conducted our survey through an online platform, telephone interviews, and face-to-face meetings with respondents. As for the online platform, we created the survey in Google Forms and shared the link online. In the case of the telephone interview, it was applied mostly to participants who were uncertain about how to complete the online survey. Due to the difficulty of finding individuals over the age of 66 on the Internet, face-to-face interviews were mostly conducted with older respondents.

  1. One of the points that you established in your conclusions are: " This study indicates a relationship between probiotic understanding 410 and education attainment. The positive correlation suggests that encouraging individuals 411 to pursue higher education may be an effective strategy for improving health outcomes. 412 Educating the public about probiotics and prebiotics, distinguishing probiotics foods and 413 probiotics supplements, and considerations when consuming probiotic products ". 

In the introduction and in the discussions this matter is not addressed, therefore it is important to establish what has been done up to now regarding education in probiotics.

Response: It has been explained in the introduction and discussion section how the role of education in probiotic product consumption has been significant and impactful.

What could we do with the information of this project? 

Response: Our study examined respondents' education levels in our study to determine whether education would impact their knowledge of probiotics. Consumer's knowledge of probiotics may influence their willingness to purchase probiotics products, which is an important consideration. Because, as of right now, Hong Kong has no legislation or standards governing the provision of information relating to the safety and efficacy of dietary supplements. Therefore, consumers are advised to determine whether or not they are suitable to consume probiotic-supplemented foods and how to consume them properly.

On the other hand, the research focuses in subjects with different ages and education backgrounds.  Based on that, is it possible to establish if people have more knowledge in probiotics based on their age.  Or is there a relation between their education background and the knowledge of probiotics?

Response: According to our analysis of the correlation between ten variables (Table 5), there was no correlation found between age and knowledge of probiotics. The results of our study suggest that higher education background is positively related to probiotic knowledge, and that enhancing education might provide people with the information they need to make wise decisions when buying probiotic products. In terms of the relationship between education background and probiotic knowledge, the next study will focus on this area.

What are the main topics that should be learned on probiotics by the people in this country. 

Response: As Hong Kong does not have any specific legislation or standards governing the provision of information regarding relevant safety and efficacy of dietary supplements, Hong Kong residents are advised to know the correct information and how to use probiotics properly by themselves to prevent side effects of misuse.

Consider the above to evaluate your conclusions.

Response: It has been improved (please refer to section “5. Conclusions”).

In addition, add in conclusions what are the next steps for this research. How will this knowledge contribute to society.  And establish what are the limits of your study.

Response: It has been improved (please refer to section “5. Conclusions”).

Round 2

Reviewer 1 Report

Comments and Suggestions for Authors

Accepted in present form